# Identifying STEDable BF_2_-Azadipyrromethene Fluorophores

**DOI:** 10.3390/molecules28031415

**Published:** 2023-02-02

**Authors:** Niamh Curtin, Massimiliano Garre, Dan Wu, Donal F. O’Shea

**Affiliations:** Department of Chemistry, RCSI, 123 St Stephen’s Green, D02 PN40 Dublin, Ireland

**Keywords:** BF_2_-azadipyrromethene, STED, super-resolution, fluorescence, far-red

## Abstract

BF_2_-azadipyrromethenes are highly versatile fluorophores used for cellular and in vivo imaging in the near-infrared and far-red regions of the spectrum. As of yet, their use in conjunction with super-resolution imaging methodologies has not been explored. In this report, a series of structurally related BF_2_-azadipyrromethenes has been examined for their suitability for use with stimulated emission depletion (STED) nanoscopy. The potential for STED imaging was initially evaluated using aqueous solutions of fluorophores as an effective predictor of fluorophore suitability. For live cell STED imaging in both 2D and 3D, several far-red emitting BF_2_-azadipyrromethenes were successfully employed. Image resolution below the diffraction limit of a confocal microscope was demonstrated through measurement of distinct intracellular features including the nuclear membrane, nuclear lamina invaginations, the endoplasmic reticulum, and vacuoles. As the STED ability of BF_2_-azadipyrromethene fluorophores has now been established, their use with this super-resolution method may be expected to increase in the future.

## 1. Introduction

The far-red region of the spectrum is useful for biological fluorescence imaging since there is a greater penetration of light at longer wavelengths, a lower phototoxicity, and cellular autofluorescence is minimal [1]. Advances into super resolution imaging in this region continue to yield a greater understanding of cellular biology with a substantial amount of research continuously ongoing [2,3]. Stimulated emission depletion (STED) nanoscopy represents one super-resolution method in which Abbe’s diffraction limit is overcome by selectively deactivating fluorophores such that the illumination area is minimized [4,5]. In STED imaging, an excitation laser (EL) is employed in conjunction with a second doughnut-shaped depletion laser (DL) [2,6]. Fluorophores are excited by the EL with a partial spatial overlapping by the DL. The DL de-excites fluorophores in the coinciding regions through a stimulated emission at the wavelength of the DL, with a higher resolution image provided from the residual non-overlapped area. With the development of the 775 nm near-infrared DL, longer wavelengths for STED imaging are now possible, potentially enabling improved resolution imaging of live cells [7].

To develop organic dyes for far-red STED imaging, rigorous physical and biocompatibility requirements must be met, including high photostability, brightness, and no light or dark toxicity [2]. Recent investigations have shown that rhodamine derivatives, π-extended-BODIPYs, and Nile red fluorophores are STED compatible when used with a 775 nm DL (Figure 1) [7,8,9,10,11,12,13,14,15,16,17]. Achievements have been made in defining subcellular structures, such as the cristae of mitochondria [8,9,10] and vimentin filaments [13] using these dye classes. With a continually increasing demand for imaging in this region some commercial dyes have also become available [7].

Once the photostability and biocompatibility criteria are met, an additional STED specific requirement remains with respect to the emission depletion [2]. Due to the high power depletion beam used, particular attention must be paid to the absorption and emission wavelengths of the fluorophore [18,19]. In general, STED fluorophores require the DL wavelength to coincide with the far-red segment of their emission spectrum, although it can be unclear what constitutes the optimal overlap [2,5,18,19]. Theoretically, as the DL wavelength approaches the emission maximum of the fluorophore, a greater depletion and resolution should be possible with lower intensity of the DL [2,19]. However, as the overlap increases, the probability of undesired anti-Stokes fluorophore excitation also increases [2,18,19,20]. When this atypical mode of excitation occurs, the fluorophore can emit its normal lower wavelength emission, which can negatively influence the STED image. In some cases, this anti-Stokes emission can be removed by lifetime gating or by subtracting the emission produced solely by the DL from the acquired data, though it would be preferential to avoid or minimize this competing photophysical process [18,20,21,22]. Moreover, as the wavelength of the DL is normally fixed and cannot be adjusted, the fluorophore parameters may need to be refined according to the DL, rather than vice versa, in order to obtain the best results. This study examines the potential of the BF_2_-azadipyrromethenes selected according to their photophysical properties across the far-red spectral region, with the objective of identifying specific derivatives that are compatible with a 775 nm DL. Although derivatives of these fluorophores have previously been used for a variety of in vitro and in vivo imaging applications, no examples of STED imaging have previously been reported. Depending on the substituents attached to the core of the fluorophore, they can emit from the far-red to the near-infrared between 640 and 1060 nm [23]. At the concentrations used for live cell imaging, they have been demonstrated excellent photostability with no cellular toxicity [23,24,25,26,27,28]. These advantageous properties encouraged our goal to identify specific examples that were suitable for STED imaging. Four previously reported and structurally related BF_2_-azadipyrromethenes **1**–**4** were chosen to investigate their STED imaging potential using a DL set at 775 nm (Figure 2) [29,30]. The core of each fluorophore was identical with different pyrrole ring substituents being either methyl, phenyl, or *p*-methoxy aryl. They were selected according to their photophysical properties spanning the far-red spectral region, exhibiting absorbance and emission maxima between 620 and 730 nm. In terms of spectral length, fluorophores **1** and **4** have the shortest and longest wavelengths respectively, with the *p*-methoxy and β-phenyl substituents of **4** contributing to the relative bathochromic shift for this derivative. Compound **2** contains four phenyl groups on the α- and β-pyrrole ring positions, while **3** has *p*-MeOC_6_H_4_ and β-methyl substituents, both of which have very similar spectral wavelengths (Figure 2).

## 2. Results and Discussion

### 2.1. In Solution Screening of STED Properties

Prior to cell imaging it was instructive to first obtain insights into the characteristics of each fluorophore in aqueous solutions as a means of pre-screening and identifying their imaging potential and impediments. In recognition that a variety of mixed hydrophilic/hydrophobic microenvironments exist within a cell, homogeneous 5 μM solutions of **1**–**4** were prepared in aqueous polysorbate 20 (PS_20_) for testing. A recording of their spectra showed **1**, at the lower end of the red spectrum, to have an absorption maximum of 623 nm and an emission of 648 nm, whereas **4**, at the upper end, has an absorption at 700 nm and an emission of 732 nm (Figure 3, Table 1). Spectrally positioned in between are **2** and **3** which have very similar absorption and emission maxima (656/684(±2) nm) but differ by their peripheral substituents (Figure 3, Table 1). Fluorescence lifetimes for **1**–**4** ranged from 2.2 to 3.9 ns with the dimethyl substituted derivatives having the longer lifetimes (Table 1, Appendix A).

In most instruments, the laser wavelengths for the depletion beam are restricted to 595, 660, or 775 nm with the latter being the one used in this study [7]. While this DL wavelength was beyond the spectral absorbance of all four fluorophores, an overlap existed to varying extents for each of their emissions (Figure 3). As such, it was informative to cross compare relative emission intensities at 775 nm versus λ_max_ for each of **1** to **4** (Table 1). This followed the trend of **4** > **3** ≈ **2** > **1** which was confirmed by their associated excitation scans for this wavelength (Table 1, Figure 3, Appendix A). These results indicate that all four should be responsive to the DL with **4** having the highest probability of anti-Stokes excitation.

As a next step, the responses of **1**–**4** to the EL, to the STED combination of EL with DL, and to the DL alone were recorded. For each fluorophore an excitation wavelength was chosen on the shorter wavelength side of the absorbance λ_max_ (Table 1). A comparison of the normalized intensity produced by the EL alone with that from the EL + DL (DL power 30%) showed a significant 50–70% reduction in intensity for **1**, **2,** and **3**, indicating that depletion was taking place as required for STED imaging (Figure 4, bars labelled (a) and (b)). However, for **4** the intensity reduction was minimal, indicating a more complex process was in progress. Similarly, the phasor plots indicated lower lifetimes for **1**–**3**, again consistent with depletion, but fluorophore **4** showed no changes (Appendix A). An understanding of **4′s** differing response to the action of the DL could be gained by measuring the response to the DL alone (Figure 4). Fluorophore **1**, which is the furthest removed spectrally from the DL showed no anti-Stokes excitation whereas **2** and **3** showed a very low but measurable fluorescence due to the action of the 775 nm DL (Figure 4, bars labelled c). These DL-induced emissions spectrally overlapped with the same λ_max_ as the EL generated emissions, though due to their low intensity would be subtractable for STED imaging (Appendix A) [20,21]. Fluorophore **4**, which is the spectrally closest to the DL wavelength showed strong anti-Stokes fluorescence, which is consistent with it having the highest overlap λ_775_/λ_max_ intensity value of 22% (Figure 4 red bar labelled (c)). Like the two previous fluorophores, this fluorescence also overlapped with the emission generated by the EL (Appendix A). Due to this strong anti-Stokes emission, it could be anticipated that this fluorophore may pose additional challenges for STED imaging with this DL (Figure 4, red bars).

### 2.2. 2D STED Imaging with 1

Having completed this initial assessment, the STED performance of fluorophores **1** through **4** was investigated in human breast cancer MDA-MB 231 cells [31]. Following a 3 h incubation with **1** comparative live cell, imaging using confocal laser scanning microscopy (CLSM) (EL λ_594_) and STED (EL λ_594_, DL λ_775_, 30% power, time gating 0.5–6 ns) was carried out (Figure 5, Appendix A). CLSM images show that **1** accumulated in several lipophilic regions including the nuclear membrane (NM), endoplasmic reticulum (ER), large vacuoles (LV), in nuclear lamina invaginations (NI) and lipid droplets (LD). When the DL was applied for STED imaging, the phasor plot showed the expected shift to shorter lifetime and a clear visual improvement was seen for each of these subcellular features.

The NM provides the nucleus with structural integrity while forming a barrier between it and the cytosol [32]. As a defined feature it was utilized to analyze the improvement in resolution for STED over CLSM (Figure 6). The full width at half maximum (FWHM) was measured at five positions of the NM for both techniques, with an average size of 307 nm for CLSM and 127 nm for STED (Figure 6 (i), intensity plot analysis for green lines). Further analysis of the NM region of interest (ROI) using image decorrelation analysis gave an image resolution of 280 nm for CLSM with a substantial image resolution improvement of 153 nm upon STED imaging [33]. This shows a remarkable improvement in resolution for the STED image, with a size measured below the diffraction limit of the confocal microscope which was recorded across repeat experiments (Appendix A). In further analysis, the graph line examination of two adjacent membranes showed that STED provided a clear separation between the two (red line) allowing for the distance between the two membranes to be measured at 250 nm (purple arrow), whereas CLSM (black line) did not provide a clear distinction between the two (Figure 6 (ii) and see Appendix A for CLSM field of view (FOV) expansion and S9 for further line analysis examples).

The ER consists of a meshwork of tubular membranes criss-crossing throughout the cytosol and was visible in both the CLSM and STED images allowing for comparison of the two methods (Figure 7) [34]. It can be seen from the graphed plots (i)–(iv) in Figure 7 of different cell regions that STED imaging outperforms CLSM with clear distinctions between ER tubules measurable for STED (red lines) compared to the CLSM (black lines). The definite separations achievable between different ER regions with STED allowed a more complete view of this complex region to be resolved by super-resolution imaging (see Appendix A for replicate experiments and other tubules separation analysis).

Another location from which **1** displayed a strong fluorescence was within LVs. LVs are a known feature of MDA-MB 231 cells that originate from the *trans*-Golgi and can comprise of several internal compartments containing secretory materials [35]. It was encouraging to observe that STED acquired images confirmed the presence of different compartments within these vacuoles, since the measured cross-sectional fluorescence intensity (yellow lines) varied between compartments (Figure 8A plots i, ii). This differed from the intensity analysis of individual LDs, which showed uniform cross sectional intensities, as would be expected of vesicles that contain triacylglycerols and cholesterol esters enclosed in a phospholipid monolayer (Figure 8B plot iii) [36].

A further interesting feature observed within the nuclear body was nuclear invaginations which have been a target for super resolution imaging [32,37,38]. NIs have been reported to develop when the nuclear membrane is pulled inwards and, if the pull is strong enough, may detach to form hollow and tube-like structures of varying size within the nucleus [32]. The exact roles of these worm-like tunnels within the nucleus remains unclear though they have been linked to intranuclear transport, cancer cell migration and metastasis [37]. Representative examples within a single cell are shown in Figure 9, some of which appear detached from the NM (expansion ii and iii) and with another (i) attached. For more STED images of NIs see Appendix A. An assessment of prolonged continuous imaging was next undertaken by acquiring FOV images of the above features every 20 s for extended periods of time. The photo-robustness of **1** was evident as no significant photobleaching occurred for 10 min, allowing for dynamic monitoring of cell regions over time (Appendix A).

### 2.3. 3D STED Imaging with 1

Next, having successfully obtained 2D STED images of **1** with various intracellular features showing an increased resolution, the acquisition of 3D images was undertaken. When imaging live in 3D, difficulties may occur due to the increased time taken for acquisition and hence increased light exposure. Challenges may arise such as movement of the cell causing distortion of the final 3D image, photobleaching of the fluorophore, and phototoxicity to the sample [39,40]. Since we are imaging using a DL in the NIR region this should help reduce the phototoxic effects that may occur to a cell when imaged at a shorter wavelength. In order to reduce the effects of photobleaching and movement whilst still maintaining a high degree of resolution, the time taken for the acquisition had to be optimized. To begin, a 3D image of a cell was acquired, however, due to the large FOV and large z-sections collected there was some movement of the cell during the acquisition. Yet, the fluorophore did not seem to have an obvious degree of photobleaching and held up well to the imaging parameters used allowing for sections of the z-stack to be analyzed (Appendix A). From this type of imaging analysis, regions such as NI still attached to the NM as well as LV containing well-defined compartments were observed (Appendix A). By restricting the number of steps and size of z-stack acquired, a 3D image of a full cell could be captured with little movement distortion. This method produced an entire cell image containing a 3D rendering of the intracellular vesicles previously mentioned (Figure 10A). In order to acquire the best 3D images, smaller z-stack of distinct FOVs containing the specific features of interest were imaged. This led to faster acquisition of the volume, the results of which can be seen in Figure 10B with (i) NI, (ii) LV, (iii) ER and LDs, and (iv) NM visible in 3D. As with the above imaging of the LV, the greater resolution can be derived from the different compartments within the vacuole. The ER contained membranes with clear separation seen between them, as well as LDs attached which appeared uniform throughout. The NM appeared consistently throughout the steps of the z-axis. Overall, through optimization of imaging parameters, 3D STED imaging with **1** was readily achieved. A resolution improvement of the intracellular features was maintained throughout imaging.

### 2.4. STED Imaging with 2 and 3

Next, an analogous series of STED and CLSM imaging experiments was conducted for fluorophores **2** and **3**. Both were incubated with MDA-MB 231 cells following the same procedure as **1,** with CLSM showing that they localized to the same subcellular regions (NM, ER, LV, NI, and LD) (Appendix A). Similar resolution improvement was observed for both **2** and **3** when the fluorophores were subjected to STED imaging (Figure 11, Appendix A). Due to the minor anti-Stokes emission detected in solution measurements for **2** and **3**, subtraction methods were employed to ensure that this did not impact the STED images. This was routinely achieved through either using the phasor plot to exclude DL generated photons or subtraction of the DL produced emission from the STED image data [20,21]. STED resolution performance with **2** was demonstrable through the analysis of the contact position in Figure 11A which was immeasurable by CLSM, whereas STED gave an intracellular distance of 270 nm (see Appendix A for analysis). Fluorophore **3** performed in a similar manner when undergoing STED imaging, as can be seen in Figure 11B with a distinct NM observed (yellow box).

Cross comparison of the FWHM of measured NM cell regions for STED images acquired using **1**, **2,** or **3** gave comparable results within an average value of 130 ± (20) nm (Appendix A). Further image decorrelation analysis gave an averaged image resolution of 140 ± (15) nm confirming that they were suitable as STED fluorophores, performing below the diffraction limit of the confocal microscope.

### 2.5. Anti-Stokes Imaging with 4

Finally, the ability of **4** to be used for STED imaging was investigated with the knowledge from its solution studies that significant challenges maybe faced due to DL-induced anti-Stokes excitation. Attempts to acquire STED cell images following incubation with **4** using EL illumination at 685 nm {collection between 705–760 nm} and the same DL power (30%) as was successfully used with **1**–**3**, produce an entirely saturated image with indistinguishable intracellular regions (Appendix A). Attribution of this result to a DL-generated anti-Stokes emission was confirmed by omitting the EL illumination from an otherwise identical experiment which also produced a saturated cell image (Appendix A). A recording of emission spectra following EL or DL excitation at 638 and 775 nm respectively gave comparable spectra with λ_max_ of 710 nm and measured fluorescence lifetimes for both excitations were identical at 3.1 ns (Figure 12A). Moreover, the intensity of the anti-Stokes emission was proportional to the power, though the image remained saturated until only 1 or 0.5% power was used (Figure 12B, Appendix A). However, DL power in excess of these values is required to produce the improved resolution effect and if saturating anti-Stoke emission is also being produced, which cannot be readily removed, **4** may be deemed non-STEDable with this DL. Therefore, to obtain super-resolution STED imagery it would be preferable to use BF_2_-azadipyromethenes **1**–**3**.

While not the focus of this work, the DL-induced emission did offer the opportunity to demonstrate that cell images can be formed using the 775 nm DL excitation at low 0.5% power [20,41]. Such an image, as shown in Figure 12C, could be termed an anti-Stokes image as it would be generated from an emission collected between wavelengths 705 and 760 nm, considerably shorter than the excitation at 775 nm. Comparison with a CLSM image of the same FOV, using excitation at 685 nm, the same intracellular regions are emissive (Figure 12D). While the CLSM produced image was clearly superior to that arising from DL excitation, overlapping of both the images showed a high degree of colocalization of emissions (Appendix A for repeat experiments). The ability to pre-detect this undesired characteristic of DL interaction with fluorophore ground state in solution experiments, as shown in Table 1 and Figure 4, could be of general value in identifying STEDable fluorophores without requiring cell experiments.

## 3. Experimental

### 3.1. Materials

Fluorophores **1**–**4** were synthesized following the literature procedures [29,30]. P_188_ and PS_20_ were purchased from Sigma Aldrich. Arklow, Ireland. MDA-MB 231 cells were purchased from ATCC. All water used was HPLC grade water purchased from Sigma Aldrich and filtered using MF-millipore, 33 mm filter.

### 3.2. General

Absorbance spectra were recorded with a Varian Cary 50 Scan ultraviolet-visible spectrometer. Emission/excitation spectra were recorded on a FluoroMax Plus spectrofluorometer. Confocal and STED images were acquired using the 3D STED Falcon (objective: Leica HC PL APO CS2 100X/1.40 oil immersion). White light laser was used to excite the fluorophores with a max power of 0.00347 mW used close to the focal plane. STED DL 775 nm with a max power of 24.7 mW (30%) was used. Images were processed using LASX, LASX Falcon (FLIM) software, and ImageJ. ImageJ plugin image decorrelation analysis was used (radius min 0, radius max 1, Nr 50, Ng 10) to calculate image resolution estimation [33].

### 3.3. Preparation of Aqueous PS_20_ Solutions

PS_20_ (100 mg) was dissolved in tetrahydrofuran (10 mL, THF). Fluorophores **1**–**4** (50 nmol) was added to the THF solution. The THF was removed under vacuum. The PS_20_/fluorophore mixture was then resuspended in water (10 mL) and sonicated for 1 min to give a final solution of PS_20_ (8.14 mM, 1% *w*/*v*)/fluorophore (5 μM). Absorbance, emission (fluorophore **1** excitation 590 nm, fluorophore **2** excitation 610 nm, fluorophore **3** excitation 630 nm and fluorophore **4** excitation 630 nm, slits excitation 1 nm, emission 2 nm) and excitation (emission peak 775 nm, slits excitation 1 nm, emission 2 nm) scans were carried out on all aqueous PS_20_/fluorophore solutions at rt.

### 3.4. STED Feasibility Study with Aqueous PS_20_/Fluorophores Solutions

Aqueous PS_20_ solutions of **1**–**4** (200 µL) were put in separate imaging wells (8 well STED compatible imaging plate) and placed on the above described Leica microscope. Using FLIM, stop condition for photon accumulation was set to 1000 photons in the brightest pixel. Illumination conditions for each fluorophore excited by white light laser (EL, **1** excited at 594 nm, **2** and **3** excited at 638 nm, and **4** excited at 685 nm) were optimized in order to have the same average photons number per laser pulse (0.25). Once optimized, the intensity/frame was measured for each fluorophore. Without changing settings, the DL (30%) was switched on and measurement repeated. To measure anti-Stokes emission, the stop condition was changed to 100 frames accumulation (to compensate the lower number of collected photons), the EL excitation was removed and the DL was kept at 30%. Emission scans produced by DL (30%) were also taken. Experiments were repeated in triplicate and average values were used.

### 3.5. Cell Imaging

MDA-MB 231 cells were seeded on an eight well chamber slide and allowed to proliferate for 24 h at 5.0% CO_2_ and 37 °C. Cells were cultured in Dulbecco’s modified Eagle medium (DMEM) supplemented with 10% fetal bovine serum (FBS), 1% penicillin/streptomycin, and 1% L-glutamate. Cells were then washed with phosphate buffered saline solution (PBS) and placed in starved cell media (DMEM supplemented with 1% penicillin/streptomycin, 1% L-glutamate and no FBS) for 24 h. This caused the number of LDs in the cell to decrease allowing for other lipophilic regions to be readily imaged [34].

### 3.6. Preparation of Aqueous P_188_ Solutions for Cell Imaging [25]

P_188_ (100 mg) was dissolved in tetrahydrofuran (10 mL, THF). Fluorophore **1**–**4** (500 nmol) was added to the THF solution. The THF was removed under vacuum. The P_188_/fluorophore mixture was then resuspended in water (10 mL) and sonicated for 1 min giving a P_188_ (1% *w*/*v*, 1.19 mM)/fluorophore (50 µM) solution. This solution was diluted 1/10 in cell media and incubated for 3 h with cells prior to imaging. Previous work in our group has shown that this procedure forms non-emissive nanoparticles that can deliver the fluorophore to cells [25]. All cells were imaged live with fluorophore **3** also imaged fixed. Fluorophore **3** was fixed in cells using 4% paraformaldehyde in PBS for 20 min.

### 3.7. Microscope Settings

Scanning speed of 200 Hz was used for all images. Fluorophore **1** was excited at 594 nm (power 0.4–1.4%), fluorophore **2** was excited at 640 nm (power 0.5–2%), fluorophore **3** was excited at 638 nm (power 0.3–3.3%), fluorophore **4** was excited at 685 nm (average power 0.4–0.9%). Fluorophores **1** and **2** were imaged using CLSM and FLIM-STED (DL 30%). TauSTED mode (τ-strength 80, denoise 100, time gate 0.5–6 ns) was used for the HyD detector to filter pixels using the phasor signature produced for each STED image [7,17]. For **2**, to ensure this did not include any photons produced from anti-Stokes, expert mode was used to set the phasor signature. STED 3D images of **1** were taken using the HyD in TauSTED mode (τ-strength 80, denoise 100, time gate 0.5–6 ns). Step size and acquisition time listed in Figure 10 and Appendix A. Fluorophore **3** was imaged using CLSM and sequentially using EL + DL (30%) and DL (30%) alone both using the HyD in TauSTED mode (τ-strength 100, denoise 50, time gate 0.5–9 ns). The image produced by DL (30 %) alone was then subtracted from the EL + DL (30%) image for further analysis. Fluorophore **4** was imaged using CLSM and EL + DL (30%). Fluorophore **4** anti-Stokes effect was imaged sequentially using DL alone (0.5%, 1%, 5%, 15%, and 30%). Max intensity inside ROI in the cytosol were measured for 0.5–30% power. Emission spectra were recorded using EL and DL alone.

## 4. Conclusions

2D and 3D live cell STED imaging was achieved using several far-red emitting BF_2_-azadipyrromethene fluorophores in conjunction with a 775 nm depletion laser. The fluorophores accumulated in lipophilic cellular regions of the endoplasmic reticulum, the nuclear membrane, nuclear invaginations, large vacuoles, and lipid droplets which allowed comparative CLSM and STED imaging to be carried out. Super-resolution performance was verified through measurement of these distinctive intracellular structures, which consistently gave values below the diffraction limit of a confocal microscope. Initial fluorophore solution measurements using the excitation and depletion lasers provided a good indication of the suitability of a fluorophore for STED imaging and may help to screen other potential STED candidates in the future. In light of the demonstrated STEDability of BF_2_-azadipyrromethene fluorophores, application of their bio-conjugated and *off*/*on* bio-responsive derivatives to continuous live cell super-resolution imaging is ongoing and will be reported on in due course.

## Figures and Tables

**Figure 1 molecules-28-01415-f001:**
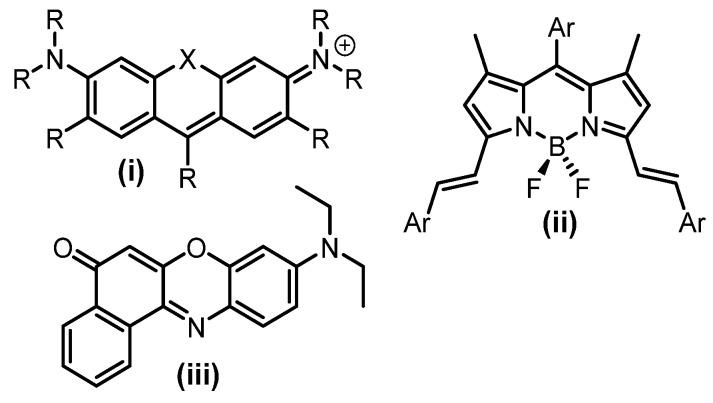
Examples of fluorophore classes used for STED imaging using DL at 775 nm. (**i**) Rhodamine derivatives, X = O, CMe_2_, SiMe_2_, (**ii**) π-extended-BODIPY, (**iii**) Nile red.

**Figure 2 molecules-28-01415-f002:**
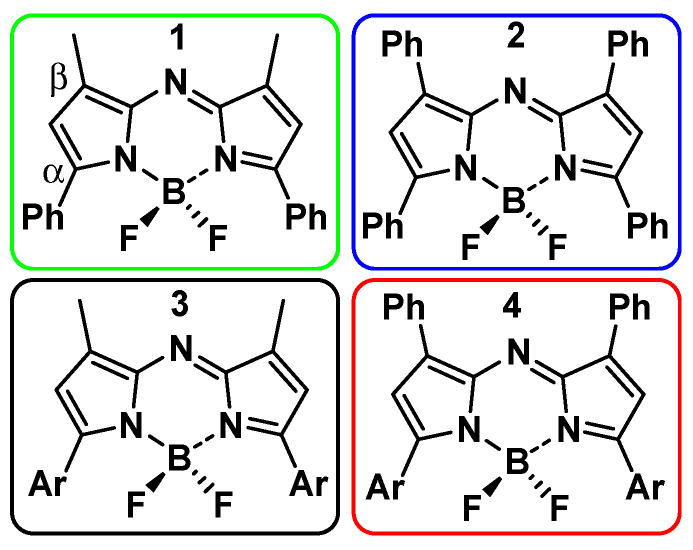
BF_2_-azadipyrromethene fluorophores **1**–**4** used in this study Ar = *p*MeOC_6_H_4_.

**Figure 3 molecules-28-01415-f003:**
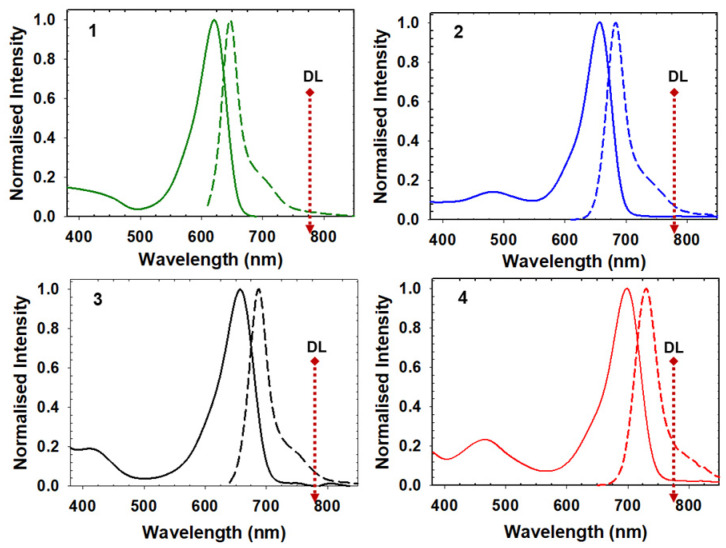
Normalized absorption and emission spectra of **1**–**4** in aqueous PS_20_ (5 μM). Dashed arrow showing the spectral position of the DL at 775 nm and the extent of overlap with the emission spectra.

**Figure 4 molecules-28-01415-f004:**
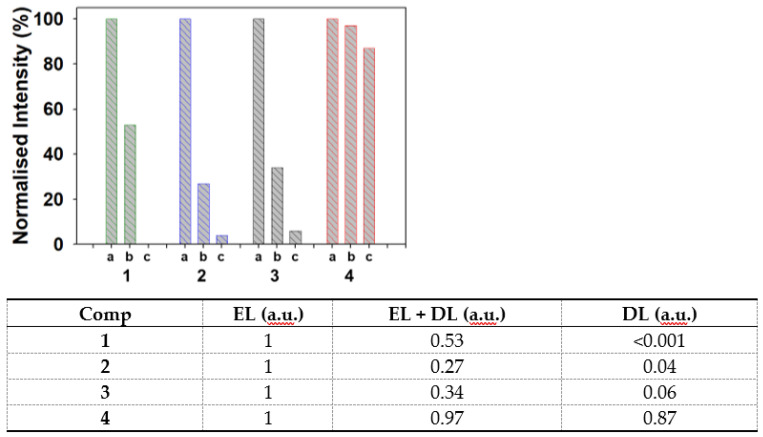
Bar graph plot and tabulated data of relative intensities per frame for **1** (green), **2** (blue), **3** (black) and **4** (red) when excited by (a) EL, (b) EL + DL and (c) DL as 5 μM aqueous PS_20_ solutions (average of triplicate experiments). EL excitations at 594 nm for **1**, 638 nm for **2** and **3**, 685 nm for **4** with power tuned to yield the same photons per laser pulse, DL power 30%.

**Figure 5 molecules-28-01415-f005:**
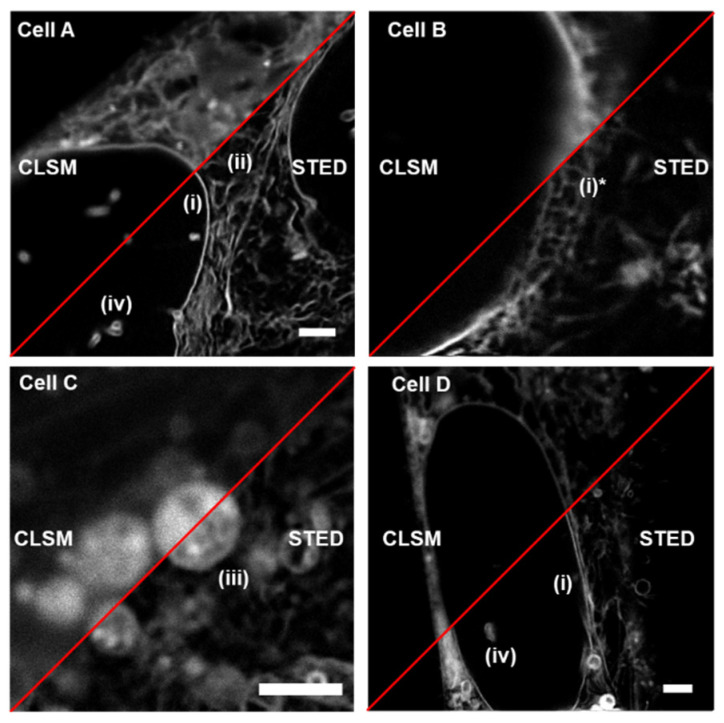
Comparative CLSM and STED imaging of **1** (5 µM) in four different live MDA-MB 231 cells. Identifiable subcellular features labelled as (**i**) NM, (**ii**) ER, (**iii**) LV (**iv**) NI, scale bars 2 µm ((**i**)* NM during mitosis).

**Figure 6 molecules-28-01415-f006:**
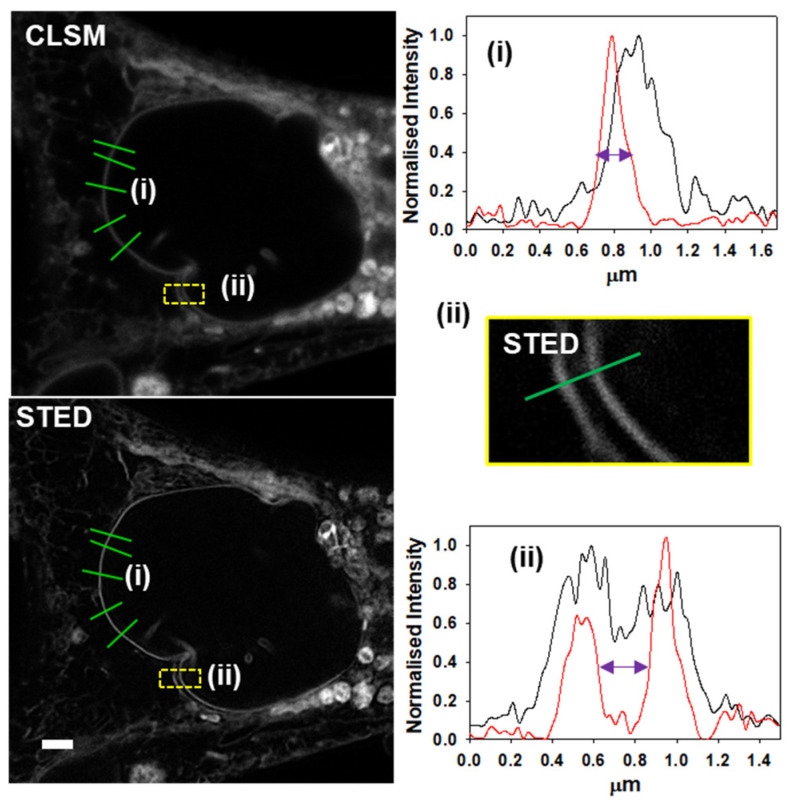
STED imaging of live MDA-MB 231 cell with **1**. CLSM and STED images of the same FOV with regions of NM intensity plot line analysis shown in green. Box (i) shows representative line analysis plot of the NM for STED (red) and CLSM (black) lines. Expansion (ii) of yellow boxed area showing STED image of adjacent membranes with box (ii) showing intensity plot analysis of green line transecting membranes for both the STED (red) and CLSM (black) lines, scale bar 2 µm.

**Figure 7 molecules-28-01415-f007:**
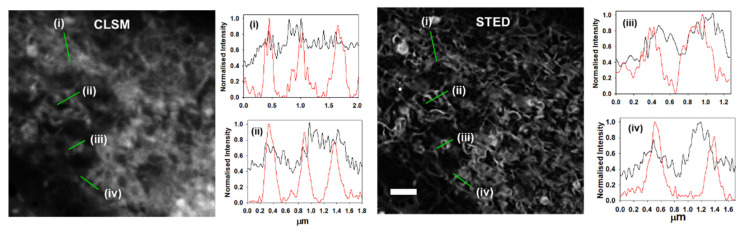
MDA-MB 231 live cell CLSM and STED imaging of the subcellular ER region. Comparative intensity plot analysis measurements showing superior performance of STED (red lines) over CLSM (black lines) in distinguishing ER tubular membranes at various positions, scale bar 5 µm.

**Figure 8 molecules-28-01415-f008:**
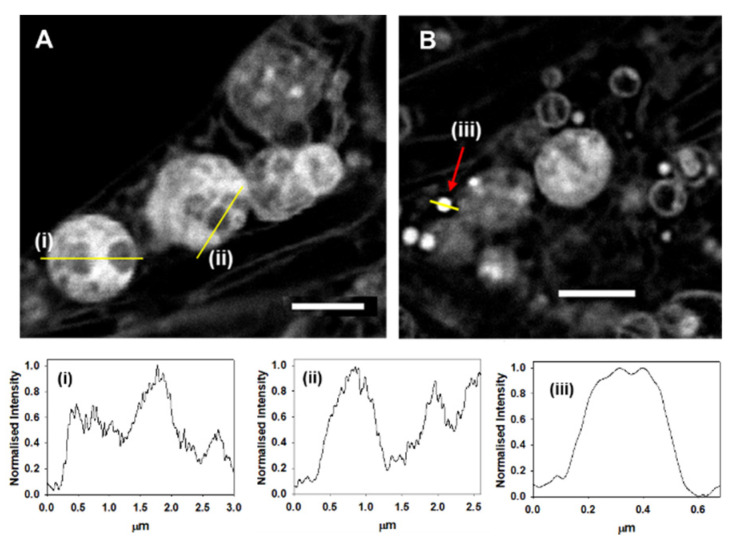
STED imaging of live MDA-MB 231 cells showing large vacuoles and lipid droplets. (**A**) Intracellular LVs with representative intensity cross-sectional line analysis (yellow lines) shown in graphs (i–ii). (**B**) Intracellular LDs with representative example indicated by red arrow for which the cross-sectional intensity line analysis (yellow line) is shown in graphs (iii). Scale bars 2 µm.

**Figure 9 molecules-28-01415-f009:**
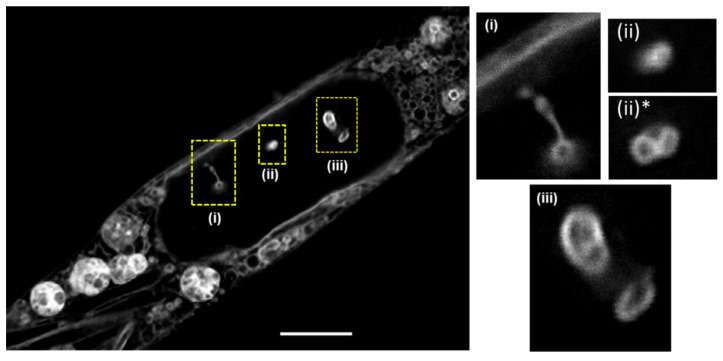
MDA-MB 231 live cell imaging of nuclear invaginations. Expansion (i) showing NI tunnel running (i) horizontal and (ii, iii) perpendicular to the image section. Expansion (ii)* at 900 nm lower focal plane than (ii) (scale bar 5 µm).

**Figure 10 molecules-28-01415-f010:**
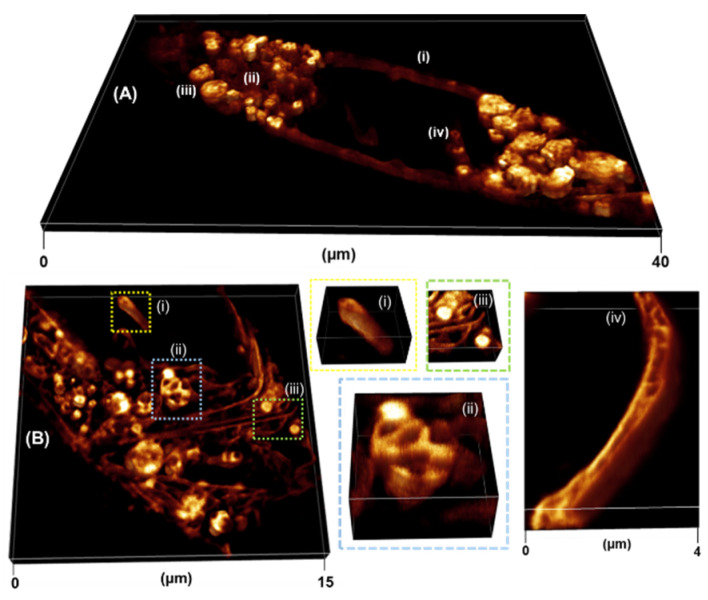
Live 3D imaging of (**A**) full cell (z-stack range 0.6 µm, 7 slices, acquisition time 25 s) with visible (i) NM (ii) ER (iii) LV and (iv) NI and (**B**) smaller volumes (i-iii) (z-stack image 1 µm, 11 slices, acquisition time 18 s) containing distinct features such as (i) NI (ii) LV, (iii) ER and LDs and (iv) NM (z-stack range 0.5 µm, 6 slices, acquisition time 10 s). Appendix A.

**Figure 11 molecules-28-01415-f011:**
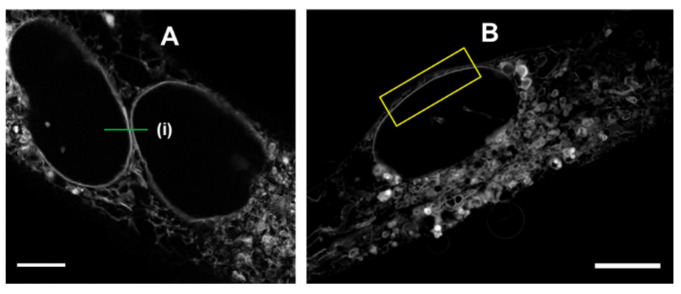
STED imaging of MDA-MB 231 cells with (**A**) **2** and (**B**) **3** (5 µM) following 3 h incubation. (**A**) Green line shows position of comparative line analysis, see Appendix A. (**B**) Yellow box indicates example area used for FWHM/ImageJ decorrelation analysis (Appendix A) scale bar 5 µm.

**Figure 12 molecules-28-01415-f012:**
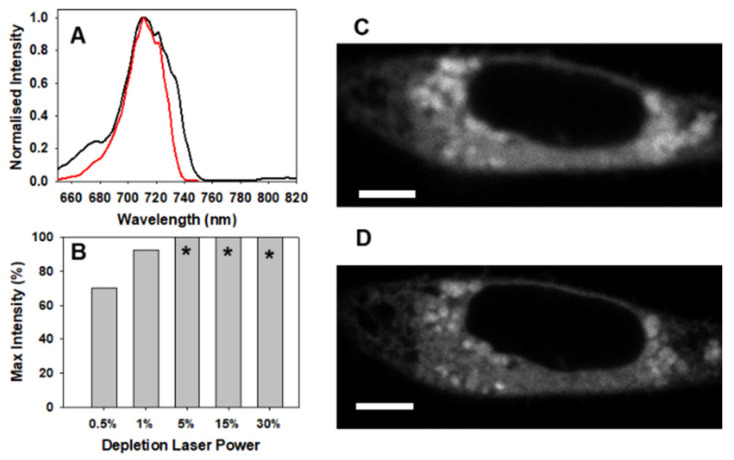
Anti-Stoke imaging of MDA-MB 231 cells with **4**. (**A**) Overlapped normalized emission spectra of **4** produced by EL (638 nm, red) and DL (775 nm, black). (**B**) Graphed illustration of % max intensity of cell images produced from DL at various power levels (* image saturated). (**C**) Cell image produced using DL (0.5%) excitation and (**D**) CLSM imaging showing emission in same intra-cellular regions, scale bars 5 μm.

**Table 1 molecules-28-01415-t001:** Photophysical characteristics of BF_2_-azadipyrromethenes **1**–**4** in aqueous PS_20_ ^a^.

Comp	λ_abs_ (nm)	λ_em_ (nm) ^b^	775 nm/λ_max_ (%) ^c^	τ_flu_ (ns)	ɸ_flu_ ^d^
**1**	623	648	3	3.6	0.41
**2**	656	682	8	2.2	0.34
**3**	656	686	9	3.9	0.44
**4**	700	732	22	3.1	0.36

^a^ 5 μM conc. ^b^ Excitation at 594, 638, 638, 685 nm for **1**–**4** respectively, slit widths excitation 1 nm, emission 2 nm. ^c^ emission intensity at 775 nm/intensity at λ_max_ × 100. ^d^ Literature values measured in organic solvents [29,30].

## Data Availability

Not applicable.

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
