# Peer review of "Identifying STEDable BF2-Azadipyrromethene Fluorophores"

_molecules, 2023, doi:10.3390/molecules28031415_

Round 1

Reviewer 1 Report

Stimulated emission depletion (STED) nanoscopy is a powerful super-resolution imaging technique in which Abbe's diffraction limit is overcome. Special attention should be paid to the selection and development of organic dyes used as STED fluorophores. The manuscript makes a substantial contribution in this field providing viable methodology for testing organic dyes as candidates for STED imaging.

In my opinion the paper generally well written and well structured, but some minor points need correction and clarification.

Section 3. Experimental directly follows Section 1. Introduction, which is unusually long. Is it possible that a header for Section 2 has been incidentally dropped?

Excitation wavelengths at 594, 638, 638, 685 nm were selected for the fluorophores, but this choice was not explained in the text. Can the excitation wavelength influence the fluorescence spectra of BF2-Azadipyrromethene?

Conclusion is very concise, some speculations about potential applications of  BF2-Azadipyrromethene in STED imaging could be useful for readers.

Author Response

thanks for the positive and constructive feedback, the following changes have been made to the manuscript; 

1.  A new "Results and Discussion" heading has been added in addition to a series of sub headings in this section.

2.  An additional line of text has been added to give more detail on the choice of excitation wavelengths.  Changing the excitation wavelengths does not influence the fluorescence spectra itself but the emission brightness will change depending upon the excitation position.

3.  the final sentence of the conclusion section has been expanded to give more details of the future potential with other BF2-azadipyrromethen fluorophores.

Reviewer 2 Report

The manuscript by Curtin et al. reports the analysis and application of BF2-azadipyrromethene fluorophores for STED microscopy. Four different derivatives are studied and characterised. The manuscript reports comprehensive and detailed spectroscopical and microscopy analysis.

Contents are well presented; data fully support conclusions, and the structure of the manuscript is clear. The proposed study may provide a useful reference and I expect the publication of the manuscript to be valuable and welcome by scientific community.

I support the publication of the manuscript as it is.

Author Response

thanks for the positive feedback, no changes to the manuscript were requested.